# Molecular Investigation of Tick-Borne Haemoparasites Isolated from Indigenous Zebu Cattle in the Tanga Region, Tanzania

**DOI:** 10.3390/ani12223171

**Published:** 2022-11-16

**Authors:** Aaron Edmond Ringo, Hezron Emanuel Nonga, Eloiza May Galon, Shengwei Ji, Mohamed Abdo Rizk, Shimaa Abd El-Salam El-Sayed, Uday Kumar Mohanta, Zhuowei Ma, Boniface Chikufenji, Thanh Thom Do, Xuenan Xuan

**Affiliations:** 1National Research Center for Protozoan Diseases, Obihiro University of Agriculture and Veterinary Medicine, Obihiro 080-8555, Hokkaido, Japan; 2Zanzibar Livestock Research Institute, Ministry of Agriculture, Irrigation, Natural Resources and Livestock, Zanzibar P.O. Box 159, Tanzania; 3Ministry of Livestock and Fisheries, Government City Mtumba, Dodoma P.O. Box 2870, Tanzania; 4Department of Internal medicine and Infectious Diseases, Faculty of Veterinary Medicine, Mansoura University, Mansoura 35516, Egypt; 5Department of Biochemistry and Chemistry of Nutrition, Faculty of Veterinary Medicine, Mansoura 35516, Egypt

**Keywords:** TBDs, indigenous zebu cattle, Tanga, Tanzania, pastoral communities, PCR

## Abstract

**Simple Summary:**

Tick-transmitted diseases are the main constraint in the development of the livestock industry in Tanzania. In this study, we investigated the occurrence of pathogens transmitted by ticks in local breeds of cattle kept under a free-range management system in the Tanga region, Tanzania. The study revealed a high infection rate of the pathogens that cause theileriosis (*Theileria parva*, *Theileria mutans* and *Theileria taurotragi*), anaplasmosis (*Anaplasma marginale*) and babesiosis (*Babesia bigemina*) in the sampled cattle. Moreover, the results show that a good number of cattle were infected with more than one pathogen. Notably, the study revealed different strains of *Babesia bigemina* and *Theileria mutans* in the population of cattle involved in the study. Furthermore, similar strains of *Theileria parva* and *Anaplasma marginale* were revealed. The information produced by this study will help policy makers to set up control strategies to contain these diseases. The control of tick-transmitted diseases will certainly improve cattle health and production. Therefore, the livelihood of the pastoralists of the Tanga region in Tanzania will be improved.

**Abstract:**

Tick-borne diseases (TBDs) are a major hindrance to livestock production in pastoral communities of Africa. Although information on tick-borne infections is necessary for setting up control measures, this information is limited in the pastoral communities of Tanzania. Therefore, this study aimed to provide an overview of the tick-borne infections in the indigenous cattle of Tanzania. A total of 250 blood samples were collected from the indigenous zebu cattle in the Tanga region, Tanzania. Then, we conducted a molecular survey using the polymerase chain reaction (PCR) and gene sequencing to detect and identify the selected tick-borne pathogens. The PCR was conducted using assays, based on *Theileria* spp. (*18S rRNA*), *Theileria parva* (*p104*), *Theileria mutans* and *T. taurotragi* (V4 region of the *18S rRNA*), *Babesia bigemina* (*RAP-1a*), *B. bovis* (*SBP-2*), *Anaplasma marginale* (heat shock protein *groEL*) and *Ehrlichia ruminantium* (*pCS20*). The PCR screening revealed an overall infection rate of (120/250, 48%) for *T. mutans*, (64/250, 25.6%) for *T. parva*, (52/250, 20.8%) for *T. taurotragi*, (33/250, 13.2%) for *B. bigemina* and (81/250, 32.4%) for *A. marginale.* Co-infections of up to four pathogens were revealed in 44.8% of the cattle samples. A sequence analysis indicated that *T. parva p104* and *A. marginale groEL* genes were conserved among the sampled animals with sequence identity values of 98.92–100% and 99.88–100%, respectively. Moreover, *the B. bigemina RAP-1a* gene and the V4 region of the *18S rRNA* of *T. mutans* genes were diverse among the sampled cattle, indicating the sequence identity values of 99.27–100% and 22.45–60.77%, respectively. The phylogenetic analyses revealed that the *T. parva* (*p104*) and *A. marginale* (*groEL*) gene sequences of this study were clustered in the same clade. In contrast, the *B. bigemina* (*RAP-1a*) and the *T. mutans* V4 region of the *18S rRNA* gene sequences appeared in the different clades. This study provides important basement data for understanding the epidemiology of tick-borne diseases and will serve as a scientific basis for planning future control strategies in the study area.

## 1. Introduction

The pastoral communities in sub-Saharan Africa practice a livestock production system that exposes cattle and other livestock to ticks and tick-borne diseases (TBDs) [1,2]. In Tanzania, the indigenous zebu cattle (*Bos indicus*), accounted for more than 96% of the total cattle population in the country. Most of these are owned by pastoralists [3]. These animals are continuously moved from one point to another in search of pastures and water. Consequently, these animals are continuously exposed to ticks and tick-transmitted pathogens. Under these circumstances, farmers are subjected to losses attributed to TBDs, which include production losses, mortality, and veterinary costs for diagnosis, treatment and control of TBDs [4,5,6]. Cattle husbandry is very important in Tanzania as more than 70% of the human population is engaged in the livestock sector [7] with a large proportion of cattle (close to 96%) owned by pastoralists and agro-pastoralists [3]. The pastoral communities mostly keep the indigenous zebu cattle and these animals are continuously exposed to different species of ticks in the pastures [8]. The main challenge of the pastoralists, apart from the shortage of pastures and water is diseases. In Tanzania, 56% of the annually reported mortalities of cattle are caused by tick-borne diseases [4,9].

Several species of hard ticks (Acari: Ixodidae) have been reported to infest cattle and other ruminants in Tanzania, including *Rhipicephalus appendiculatus*, *R. microplus*, *R. pulchellus*, *R. pravus*, *R. punctatus*, *R. composites*, *R. sanguineus*, *R. evertsi evertsi*, *Amblyomma variegatum*, *Am. gemma*, *Am. lepidum*, *Hyalomma rufipes*, *H. truncatum* and *H. albiparmatum* [10,11,12,13]. These tick species transmit various tick-borne pathogens (TBPs) which cause highly pathogenic and economically important TBDs in cattle in Tanzania, i.e., east coast fever (ECF), babesiosis, anaplasmosis and ehrlichiosis [14]. Moreover, the less pathogenic but important TBPs in cattle reported in Tanzania include *Theileria mutans* and *T. taurotragi* [15,16].

*Theileria parva* is an intracellular protozoan parasite transmitted by the three-host *tick species R. appendiculatus* and causes ECF, which is an acute, lymphoproliferative disease accompanied by a high fever, anorexia, and enlarged lymph nodes, and usually leads to death in cattle [17]. East coast fever is distributed in Eastern, Central and Southern Africa, corresponding with the range of its vector *R. appendiculatus*. The distribution mainly relies on the vector dynamics, grazing system, host susceptibility and tick control practices. Another TBP, *Theileria mutans*, was previously considered to cause a benign form of theileriosis, but pathogenic strains of *T. mutans* have been reported to cause disease in cattle in East Africa [18]. The clinical symptoms of *T. mutans* infections in cattle include severe anemia, icterus, enlarged lymph nodes, and weight loss, and sometimes death [19]. These clinical symptoms can be confused with the mild form of *T. parva* infections in cattle. *T. mutans* parasites are transmitted by three-host ticks of the genus *Amblyomma,* and the African buffalo (*Syncerus caffer*) is a wildlife reservoir [20]. In contrast, *T. taurotragi* is known for causing a mild form of theileriosis and has been incriminated as a causal agent of bovine cerebral theileriosis (turning sickness) [21]. Other clinical signs manifested by a *T. taurotragi* infection includes a mild febrile reaction and slight enlargement of the superficial lymph nodes. *T. taurotragi* is transmitted by a three-host tick *R. appendiculatus*, the same vector reported to transmit *T. parva* in Tanzania. Apart from cattle, the other reported hosts of *T. taurotragi* are elands, bushbuck, and buffalo [20,22,23,24].

Bovine babesiosis is caused by piroplasmid protozoa *B. bigemina* and *B. bovis* in Tanzania [16,25]. Of the two species, *B. bovis* is more virulent and occasionally causes a severe form of babesiosis with higher mortality rate in the country [26]. The two pathogens are transmitted by a one-host tick *R. microplus* in Tanzania [27]. Bovine anaplasmosis, also known as gall sickness in cattle, is caused by *A. marginale* in Tanzania [28]. The parasite is an obligate intracellular rickettsia that causes fever, progressive anemia, and icterus in cattle. The principal vector of this bacteria in Tanzania is *R. microplus*, but it can be transmitted by other blood-sucking insects and mechanically by blood-contaminated fomites [25]. Ehrlichiosis is caused by an intracellular *Ricketssia* parasite *E. ruminantium*, and is transmitted by a three-host tick *Am. variegatum* in Tanzania [29]. The disease is characterized by pyrexia, anorexia, dullness and nervous signs.

Despite the importance of livestock TBDs in the country, information on the epidemiology of these diseases and the genetic composition of the causative agents in Tanzania is limited. Therefore, this study aimed to investigate the occurrence and genetic diversity of bovine tick-borne pathogens of veterinary significance on the northeastern coast of Tanzania.

## 2. Materials and Methods

### 2.1. Study Area

This study was conducted in the Tanga region, which is located on the northeast coast of Tanzania. The region covers 26,680 km^2^, about 2% of the size of Tanzania. The Tanga region lies between latitude 4.965088° S and 5.5743° S and longitude 38.2744° E and 38.7787° E (Figure 1). The sample collection sites are located in the Handeni district, where the topography is characterized by low lands and rises to the average elevation of 696 m. The district is characterized by short vegetation towards the inlands which also becomes semi-arid and dry. Handeni has an annual precipitation of 600 mm which begins from April to May, followed by a dry season to December. From January to March, the area is cool with some occasional precipitation. The average temperature is between 26 °C to 32 °C. The district is composed of 21 wards.

### 2.2. Sample Collection and DNA Extraction

A total of 250 blood samples were collected randomly from apparently healthy indigenous zebu cattle in five herds, located in different wards of the Handeni district: Kwamsisi (*n* = 51), Komkonga (*n* = 49), Kabuku (*n* = 50), Kwamatuku (*n* = 50), and Kwachaga (*n* = 50) (Figure 1). Of the sampled animals, 184 were female cattle and 66 were males. Samples were collected between May and June 2019. The five herds were managed under extensive grazing on communal land by pastoralists. Each herd consisted of approximately two hundred cattle. For each animal, approximately 4 mL of blood was collected from the jugular vein, using a sterile vacutainer needle and collected in vacutainer tubes (BD Biosciences, Franklin Lakes, NJ, USA) coated with ethylenediaminetetraacetic acid (EDTA). We targeted cattle of one and a half years old and above. The animal’s age was obtained through the farmer’s records and by using the horns of the sampled animals. Samples were temporarily stored in cool boxes in the field before being moved to the laboratory for refrigeration at 4 °C. DNA was extracted from 200 µL of whole blood, using QIAamp DNA Blood Mini Kit (Qiagen, Hilden, Germany) following the manufacturer’s protocol, and stored at −80 °C.

### 2.3. Ethical Statement

The permission to sample the animals was granted by the Ministry of Livestock and Fisheries-Tanzania, and all of the farmers were informed about the importance of the study and gave their consent, under the condition that blood should be drawn by experienced veterinarians and proper restraint procedures should be applied to avoid any injuries to their animals. All of the required procedures for the sample collection were carried out, based on the ethical guidelines for the use of animal samples, permitted by Obihiro University of Agriculture and Veterinary Medicine, Hokkaido, Japan (Animal experiment permit no. 20-128).

### 2.4. Molecular Detection of the Tick-Borne Pathogens

DNA samples were screened for *B. bigemina*, *B. bovis*, *A. marginale* and *E. ruminantium* by the nested PCR with species-specific primer assays (Table 1). *Theileria* spp. Were screened using the genus-specific 18S rRNA primers. Thereafter, the *Theileria* genus positive samples were screened using the *Theileria* species-specific primer assays. The PCR assays were performed using a total of 10 µL volume containing 0.5 µM of each primer, 2 µL of 10x standard *Taq* buffer, 0.5 µL of deoxynucleotide triphosphates mix (dNTPs), 0.05 µL *Taq* DNA polymerase (New England Biolabs, Hitchin, UK), 1.5 µL of the purified DNA sample and 4.95 µL of UltraPure™ DNase/RNase-free distilled water (Invitrogen, MA, USA). The PCR-positive samples from a previous study [30] were used as positive controls in this study, while the UltraPure™ distilled water was used as a negative control. The products were run in a thermal cycler (VeritiTM Applied Biosystems, MA, USA). The PCR thermal cycling conditions in this study were obtained from the previous studies (Table 1). The PCR products were electrophoresed on a 1.5% agarose gel, followed by staining on ethidium bromide and viewed on a UV transilluminator.

### 2.5. Sequencing of the PCR-Positive Samples

We randomly selected 5–10 positive samples (approximately 10%) from each detected pathogen for sequencing. The amplicons were extracted from the agarose gel by the QIAquick Gel Extraction Kit (Qiagen, Hilden, Germany). The concentration of each extracted PCR product was checked with a NanoDrop 2000 spectrophotometer (Thermofisher, Waltham, WA, USA). The Big Dye terminator cycle sequencing kit (Applied Biosystems, MA, USA) and ABI PRISM 3130xl Genetic Analyzer (Applied Biosystems, MA, USA) were used to perform all sequencing assays. The produced sequence reads were checked and analyzed with the web-based software named Mixed Sequence Reader (MSR) (http://MSR.cs.nthu.edu.tw/, accessed on 27 September 2022), to clean them for heterozygous base calling, such as indels, tandem repeats and single nucleotide polymorphism (SNIP). The sequence reads for each sample were trimmed and assembled by SeqMan Pro software from DNASTAR Lasergene, to obtain the consensus sequence for each selected positive sample. The GenBank BLASTn analysis was used to check the sequence identities with sequences previously deposited in the GenBank database. The sequences of the same pathogens were checked for the percentage identity by using an online software tool called Sequence Identity and Similarity (SIAS).

### 2.6. Phylogenetic Analysis

The gene sequences of *T. parva* (*p104*), *B. bigemina* (*RAP-1a*), *A. marginale* (*groEL*), and *T. mutans* and *T. taurotragi* (V4 region of the *18S rRNA*), obtained in this study and those deposited in the GenBank database reported from previous studies, were used for the phylogenetic analysis, using MEGA version XI [38]. The maximum likelihood and neighbor-joining methods were used with the bootstrap set at 1000 replicates.

### 2.7. Nucleotide Sequence Accession Numbers

The gene sequences generated from the present study were deposited in the Genbank database of the National Center for Biotechnology Information (NCBI) using BankIt for genomic DNA sequences and the ribosomal RNA submission portal (submit.ncbi.nlm.nih.gov/subs/genbank/, accessed on 27 September 2022) for the ribosomal RNA sequences. The GenBank accession numbers assigned to the sequences of this study are as follows: OP390271–OP390278 for *T. parva p104*, OP390279–OP390284 for *B. bigemina RAP-1a*, OP414689–OP414693 for *A. marginale groEL*, OP379365–OP379368 for *T. mutans* V4 region of the 18S rRNA and OP380376–OP380382 for *T. taurotragi* V4 region of the 18S rRNA.

### 2.8. Statistical Analysis

The prevalence of the detected pathogens was statistically analyzed by Pearson’s chi-square (Χ^2^) and Fisher’s exact test and the significance of the co-infections was determined by the odds-ratio calculation using MedCalc software. A *p*-value < 0.05 was considered statistically significant.

## 3. Results

### 3.1. Overall Infection Rate

Out of the 250 cattle screened by PCR, 218 cattle (87.2%) were positive for at least one of the five detected pathogens, while 32 cattle (12.8%) were not infected by any of the screened pathogens (Table 2). The detected pathogens in this study were *T. mutans* (48%; 120/250), *A. marginale* (32.4%; 81/250), *T. parva* (25.6%; 64/250), *T. taurotragi* (20.8%; 52/250) and *B. bigemina* (13.2%; 33/250) (Table 3). The overall prevalence, based on location is indicated in Table 3 and there were no significant differences in infection rates, based on the five locations. Based on animal’s sex, 58 males (87.9%) and 153 females (83.2%) were infected by one or more of the screened pathogens (Table 4). There were no significant differences observed, based on sex (Table 4). *B. bovis* and *E. ruminantium* were not detected in this study.

### 3.2. Co-Infection Analysis

The proportion of cattle in which two or more pathogen species were detected is referred to as co-infections. Out of 250 cattle in this study, 112 cattle (44.8%) were found to be simultaneously infected with two or more pathogens. Overall, 48 concurrent infections were observed in this study (Table 2). The pathogen co-infections ranged from double to quadruple (Table 3). The double co-infection (79.5%; 89/112) contributed the majority of the pathogens involved in the co-infections, followed by the triple co-infection (18.8%; 21/112) and there were two cases (1.8%; 2/112) of quadruple co-infections (Table 5). The most frequent combinations of co-infection were *T. mutans* + *A. marginale* (18), followed by *T. mutans* + *T. parva* (17) and *T. mutans* + *T. taurotragi* (14) (Table 2). The co-infections, based on location, showed that Kwachaga had the highest prevalence (54%; 27/50), followed by Kwamsisi (50.9%; 26/51), Komkonga (42.9%; 21/49), while Kabuku and Kwamatuku both had 19 (38%) (Table 2). There were no significant differences in co-infection rates, based on location.

### 3.3. Comparative Gene Sequence Analyses

The sequences of each detected pathogen were compared for the nucleotide identity among themselves and with the corresponding sequences isolated previously in Tanzania or neighboring countries. The percent nucleotide identities of eight *T. parva p104* gene sequences (OP390271–OP390278) ranged from 98.92–100% with each other. In addition, these sequences showed percent identity values of 98.20–99.28%, with sequences MG700532, MG700532, MG210825 and MN807320 obtained previously in Tanzania. Meanwhile, the shared percent nucleotide identity values of six *B. bigemina RAP-1a* gene sequences (OP390279–OP390284) obtained in this study ranged from 99.27–100%. These sequences also showed an identity value of 99.51% with sequences MG210822–MG210824, obtained from previous studies conducted in Tanzania. Interestingly, these sequences were more identical (99.76%) with sequences MG426199 and KP347559 from Uganda and Kenya, respectively. For *A. marginale*, the nucleotide identity values of five *groEL* gene sequences (OP414689–OP414693) showed high identity values of 99.88% to 100%. However, these sequences showed a 99.10% identity value with sequences KY523020–KY523026 from Uganda. Furthermore, sequences (OP379365–OP379368) of the 18S rRNA gene of *T. mutans* showed low percentage identity values ranging from 22.45% to 60.77% among them, however, they showed higher identity values of up to 98.85% with sequences MN726645, MN726648, MN726650 and MG755217 isolated previously in Tanzania. Finally, the 18S rRNA gene sequences (OP380376–OP380382) of *T. taurotragi* showed a high diversity between them with identity values of 21.81% to 94.65%. Surprisingly, these sequences showed a high identity value (100%) with sequences MN726630, MN726631, MN726635 and MG755215 isolated in previous studies in Tanzania.

### 3.4. Phylogenetic Analysis

In this study, the phylogenetic trees of *T. parva*, *B. bigemina*, *A. marginale* and *T. mutans* were constructed, based on their respective genes (*p104*, *RAP-1a*, *groEL*, and V4 region of *18S rRNA*), together with sequences extracted from the NCBI GenBank database. All sequences of the *p104* gene of *T. parva* were clustered together on a phylogenetic tree (Figure 2). Notably, sequence MN810050 from Uganda and KP347564 from Kenya, showed a close relationship with the clade that contains sequences of this study (Figure 2). For *B. bigemina*, the *RAP-1a* gene sequences OP390281 and OP390279 of this study appeared in the same clade, together with sequences KY484520 from Indonesia, MK481015 from South Africa, and MG210822–MG210823 from Tanzania. The other three sequences OP390280, OP390283 and OP390282 of this study appeared in a different clade with sequences MN870655 from Egypt, MN807306 from Tanzania and MG426198 from Uganda (Figure 3). Moreover, all *A. marginale* sequences OP414690–OP414693 of this study appeared in a single clade on a phylogenetic tree, plus a sequence KC113455 from the Philippines (Figure 4). For *T. mutans*, the V4 region of the 18S rRNA gene sequences OP379365 and OP379366 of this study formed a clade of their own, while the other OP379367 appeared isolated from the other sequences in the tree. Moreover, sequence OP379368 formed another clade with sequence MN726646 from Tanzania and AF078815 from Kenya (Figure 5).

## 4. Discussion

This study demonstrates a high prevalence of tick-borne pathogen infections among indigenous zebu cattle in the pastoral community of the Tanga region, Tanzania. However, some of the detected pathogens showed some degrees of diversity. Our findings revealed that *T. mutans*, *T. parva*, *T. taurotragi*, *B. bigemina* and *A. marginale* were present in the sampled animals. Remarkably, high rates of co-infections were observed in the sampled cattle.

The proportion of cattle infected with *T. mutans* in this study was higher, compared to other detected pathogens. The prevalence of *T. mutans*, a pathogen transmitted by the *Am. variegatum* tick is in agreement with previous studies conducted in Tanzania [14,16,30] and the neighboring countries of Malawi [39], Zambia [40,41], Uganda [2] and Kenya [42]. The higher prevalence of this pathogen in this study can be explained by the fact that in endemic areas, cattle can acquire these pathogens and carry them for long periods without manifesting clinical symptoms [43]. Presumably, [44] reported that calves acquire the infection when young (five to six months) in endemic areas and they remain lifelong carriers of the disease, whereby they continuously infect ticks and subsequently cause new infections in cattle. The other possibility could be that the *Am. variegatum* ticks reported in Tanzania [12,13,25] could be widely distributed and efficiently transmitting the pathogen.

Importantly, some strains of *T. mutans* reported to be pathogenic in East Africa, are of economic importance to the farmers and require proper intervention as they can cause serious diseases in exotic breeds or newly introduced naive cattle. Equally important, is that the symptoms manifested by the *T. mutans* infections are somehow similar to *T. parva* infections, therefore, clinically, these two infections can be confused in the field. The phylogenetic tree analysis for the V4 region of the *18S rRNA* gene sequences of *T. mutans* shows that the sequences were clustered in different clades, which implies that different genotypes of *T. mutans* are circulating in a population of cattle in the study area. The variants of the *18S rRNA* gene shown in this study are in consistent to those shown in previous studies conducted in Uganda [2], South Africa [20] and Malawi [39], indicating a wide genotypic diversity of *T. mutans* in Eastern and Southern Africa.

About a quarter of the sampled cattle were positive for *T. parva* in this study. This pathogen is the causative agent of ECF in cattle and is the most economically important protozoan tick-borne pathogen in Tanzania [45]. Notably, the prevalence of *T. parva* in this study is relatively lower, compared to studies conducted on the nearby islands of Pemba [16] and Zanzibar [30], in Tanzania. However, these findings are consistent with the other study conducted in the regions of Mara, Singida and Mbeya in Tanzania [13]. Furthermore, similar results were reported in the neighboring countries of Uganda [2,46] and Zambia [41]. The lower infection rates of *T. parva* in this study could be attributed to the fact that the indigenous zebu cattle (kept under an extensive management system) are fairly resistant to *T. parva* infections [47]. This is due to the continuous exposure to this pathogen from the early stages of their lives; therefore, cattle develop an immunity against the *T. parva* infections. These animals, which are mostly under the traditional management system, tend to develop a state of endemic stability, by developing antibodies against *T. parva* infections which protect them from any further reinfection [48]. Moreover, the other possibility could be the reduced abundance and poor distribution of its vector *R. appendiculatus,* in the study area, which is dry and semi-arid with very low precipitation. Tanzania, as any other tropical country, has been affected by climate change in various agroecological zones, culminating in a change in rainfall patterns, reduced rainfall and the emergence of drought in different zones [49]. Drought has been hypothesized to reduce the abundance and distribution of tick vectors, such as *R. appendiculatus* [47,50]. The phylogenetic tree analysis shows that the *T. parva p104* gene sequences of this study were clustered in a single clade, suggesting that the *p104* gene is conserved in the sampled cattle. The sequences of this study show a close similarity with other sequences previously reported from Tanzania and neighboring Uganda (Figure 2), suggesting that the *T. parva* isolates from this study belong to the same genotype (Figure 2). The similarity could be attributed to the free-range management system practiced in the area, which enable the interactions between ticks and different groups of cattle. The results are not consistent to the previous study conducted in Burundi [51]. However, a study conducted in Uganda [52], showed variations from isolates which could be due to the study conducted at the interface with the national park, therefore there could be a mix of the buffalo derived strains with those from cattle.

*Theileria taurotragi* is a piroplasm parasite that causes benign theileriosis in cattle. This study reported a low infection rate of *T. taurotragi*. However, *T. taurotragi* and *T. parva* are transmitted by the same vector *R. appendiculatus*. Coincidentally, the two pathogens in this study had a relatively lower infection rate, which can be presumably related to the low abundance and distribution of their vector. The comparison of the *T. taurotragi* prevalence with other studies conducted in Tanzania shows that it was lower, compared to previous studies conducted on the nearby islands of Pemba [16] and Zanzibar [30], which may suggest that the sampling sites of the present study could not provide the same favorable environment for the multiplication of the pathogen and its vector. More importantly, *T. taurotragi* has been associated with neurological signs in indigenous zebu cattle in East, Central and Southern Africa [15,18]. This implies that the pathogen is of economic importance in this region.

Babesiosis is an important disease in cattle and other ruminants. *Rhipicephalus microplus* is currently the main vector and efficiently transmits bovine *Babesia* spp., while *R. (Boophilus) decoloratus* was previously the reported vector of *Babesia* spp. in the country [27]. Notably, the prevalence of *B. bigemina* in this study is consistent with the previous studies conducted in Tanzania [16,29,30]. Moreover, similar results were reported in the neighboring countries by [53] in Uganda, [54] in Kenya and [39] in Malawi. The consistency in the prevalence in the region could be due to the emergence of the *R. microplus* tick, which is vastly distributed,3 due to its high ability to reproduce and its ability to adapt to different climates [55]. The phylogenetic tree analyses show that sequences of the *RAP-1a* gene of this study appeared in different clades which suggests that different isolates of *B. bigemina* are circulating in a population of cattle in the study area. Similar findings were reported previously in Tanzania [30,51], Uganda [53] and Kenya [54], which implies that different strains of *B. bigemina* are circulating in the region.

Bovine anaplasmosis is an important tick-borne pathogen of cattle and is more devastating in adult exotic breeds [44]. In this study, we report a slightly higher prevalence than the previous reports in Tanzania [14,16,30]. Similar findings have been reported in the neighboring countries of Burundi [56], Malawi [39] and Zambia [41]. The higher prevalence of *A. marginale* in this study could presumably be due to the wide distribution of *R. microplus* in the coastal areas of Tanzania [27]. The other possibility for a higher prevalence of this pathogen can be due to the ability of the indigenous zebu cattle to develop a resistance against TBDs. This has been a protective mechanism used by the indigenous zebu cattle to survive the challenge of tick-borne infections. Therefore, when infected with *A. marginale*, they have a fast mechanism of seroconversion and developing immunity against the pathogen, hence, they can continuously be infected without showing clinical symptoms [57]. The phylogenetic tree analysis of the *groEL* gene sequences revealed that sequences of this study were clustered in the same clade, suggesting that similar genotypes of *A. marginale* are present and circulating in the population of cattle in the study area. The analyses reveal a consistency with the findings obtained in Benin [58], which suggest that in some areas of sub-Saharan Africa, this gene is conserved. However, in Uganda, a study conducted in Kasese district at the interface with the Queen Elizabeth National Park [52] reported some degree of diversity of this gene, which could be due to the interaction of wild animals and cattle in the study site, and therefore some strains could originate from the wild animals.

The prevalence of *T. parva* in this study, was lower, compared to studies conducted in Kenya [59,60] and Uganda [61]. The lower prevalence of *T. parva* in this study, are based on the fact that east coast fever is endemic in East, Central and Southern Africa, therefore cattle, especially those managed under a free-range management system, are exposed to *T. parva* infections in the early stages of their lives and develop the antibodies against the infection which protects them against further infection. Thus, in most cases, animals will have the persistent antibodies showing exposure to infection. This explains the higher prevalence shown by the serological tests, compared to the PCR test of this study. The same explanation can be used on the prevalence of *B. bigemina,* which showed a lower prevalence, compared to a study conducted in Tanzania [14] and Kenya [62]. Moreover, the prevalence of *A. marginale* in this study, was lower, compared to previous studies conducted in Kenya [62] and Uganda [63]. this can be attributed to the fact that both serological studies used indigenous breeds of cattle which are tolerant to the tick-borne infection, following early exposure to these pathogens, and they then developed antibodies. The antibodies are detected by the serological tests but not the PCR. The prevalence of *T. mutans* in this study, was higher, compared to studies conducted in Kenya [59,64], the higher prevalence of this study, compared to the two serological studies could be caused by the use of a high number of calves in the two serological studies, which suggest that the calves were not yet exposed to *T. mutans* infection. Moreover, the maternal antibodies in most of the calves might have worn out at the time of sampling, hence the serological tests appeared to have shown a lower prevalence, compared to the PCR results of this study.

Remarkably, this study reports a high prevalence of co-infections, implying that a significant number of cattle were concurrently infected with multiple parasite infections at the time of sampling. Concisely, the co-infections scenario in the endemic areas has been documented to increase or decrease the pathogenicity of the infections, depending on the pattern of the parasites involved in the co-infections [65]. This heterologous reactivity plays a role in the patterns of morbidity and mortality of the diseases. Therefore, in parasitic diseases, the outcome of the co-infection can either be protective to the host or may lead to severe infections to the host, depending on the pathogens involved in the co-infections. Altogether, the interaction of tick-borne parasites during co-infections in the indigenous zebu cattle in endemic areas, suggests that there can be ecological and epidemiological mechanisms to survive the challenges of the TBDs [66,67]. However, the higher prevalence of co-infections reported in this study suggests that a poor management system is imposed by the pastoralists on their animals in this study.

## 5. Conclusions

This study revealed a high prevalence of tick-borne pathogens in indigenous zebu cattle of the Tanga region in Tanzania. Additionally, the *RAP-1a* and the V4 region of the *18S rRNA* genes of *B. bigemina* and *T. mutans*, respectively, showed that different strains of the two pathogens are circulating in the study site. On the contrary, the *p104* and *groEL* genes of *T. parva* and *A. marginale*, respectively, revealed that similar strains of the two pathogens are circulating among a population of cattle in the transhuman pastoral system of the Tanga region. The study also showed that co-infections were more common than single infections, which implies that the interaction of the pathogens involved might have an effect on the pattern of clinical symptom manifestation and therefore, complicate the diagnosis of the diseases. The epidemiological data produced in this study provide significant information on tick-borne diseases in the area and will serve as a scientific basis for planning future control strategies.

## Figures and Tables

**Figure 1 animals-12-03171-f001:**
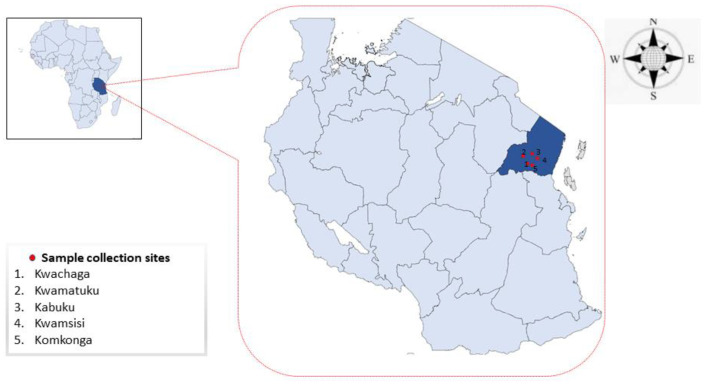
Map of Tanzania showing the sample collection sites.

**Figure 2 animals-12-03171-f002:**
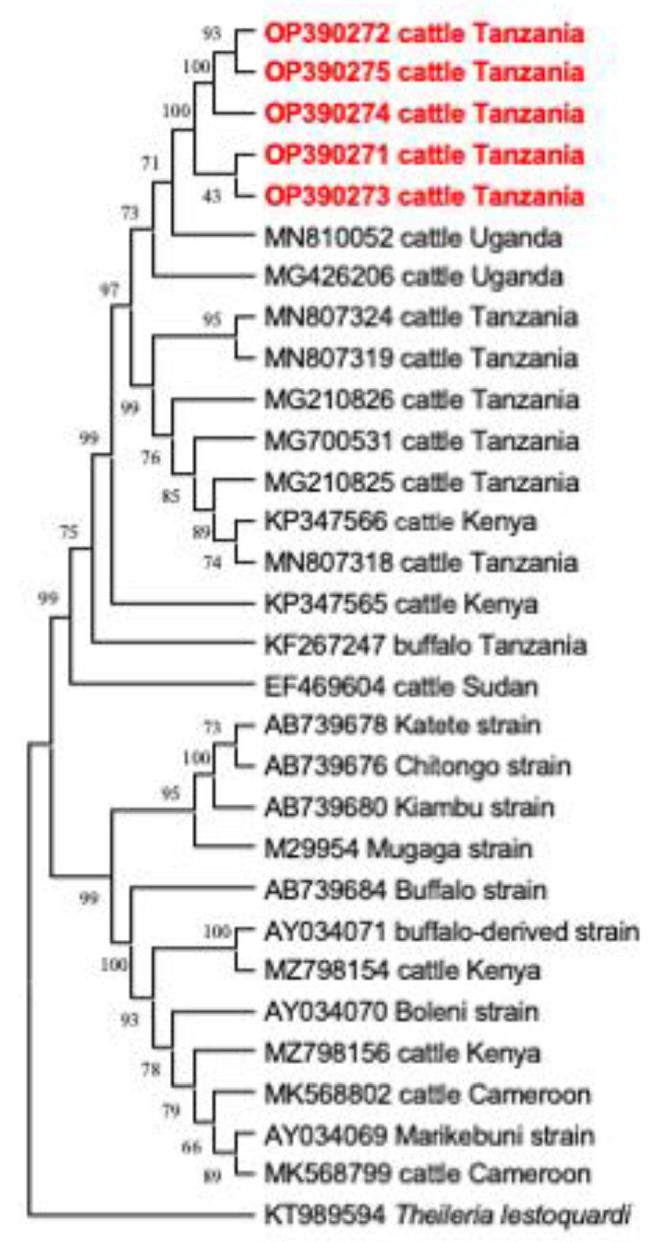
The phylogenetic analysis of *Theileria parva* identified in this study, based on the (*p104)* gene sequences. The tree was constructed by MEGA version 11 using the maximum likelihood method. The numbers at nodes represent the percentage occurrence of the clade in 1000 bootstrap replications of the data. Sequences from this study are shown in red. *Theileria lestoquardi* (KT989594) was used as an outgroup.

**Figure 3 animals-12-03171-f003:**
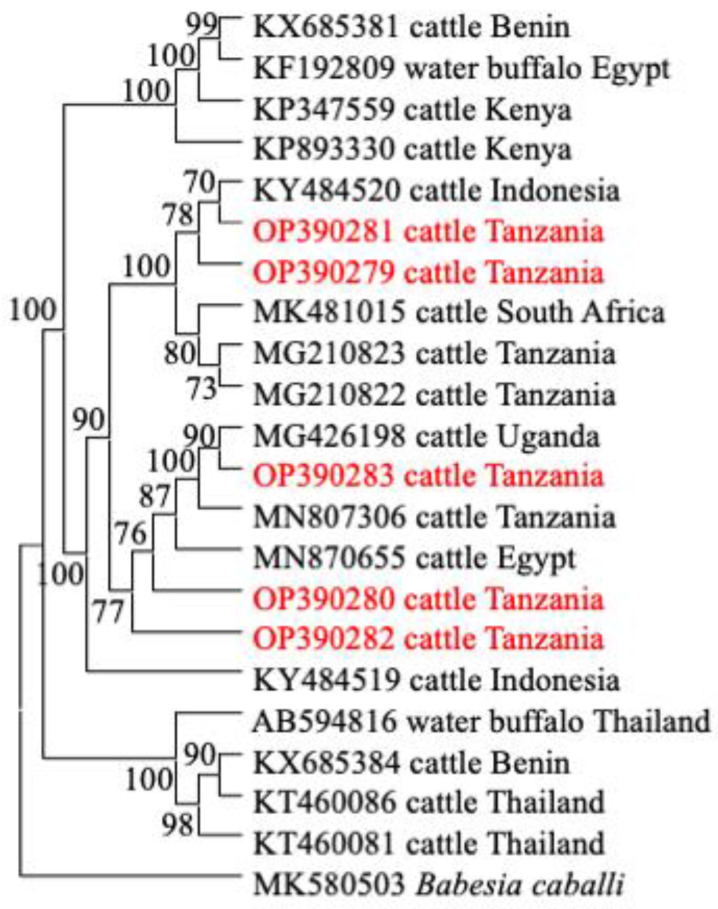
The phylogenetic analysis of *Babesia bigemina* identified in this study, based on the (*RAP-1a*) gene. The tree was constructed by MEGA version 11 using the maximum likelihood method, the confidence of occurrence of the nodes was assessed by bootstrap in 1000 replications. The sequences of this study are shown in red. *Babesia caballi* (MK580503) was used as an outgroup.

**Figure 4 animals-12-03171-f004:**
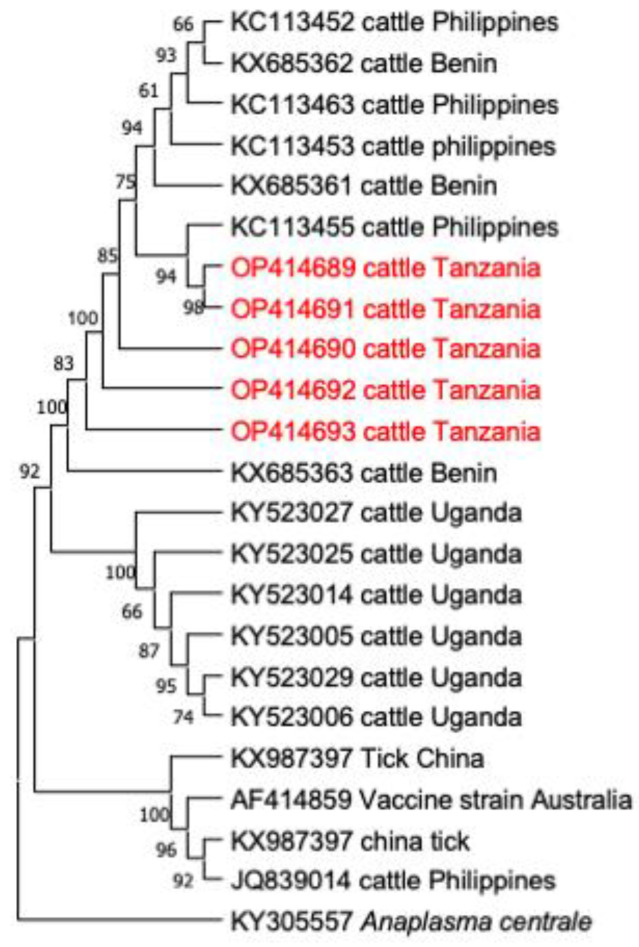
The phylogenetic analysis of *Anaplasma marginale* identified in this study, based on the (*groEL*) gene. The tree was constructed by MEGA version 11 using the maximum likelihood method, the confidence of occurrence of the nodes was assessed by bootstrap in 1000 replications. The sequences of this study are shown in red. *Anaplasma centrale* (KY305557) was used as an outgroup.

**Figure 5 animals-12-03171-f005:**
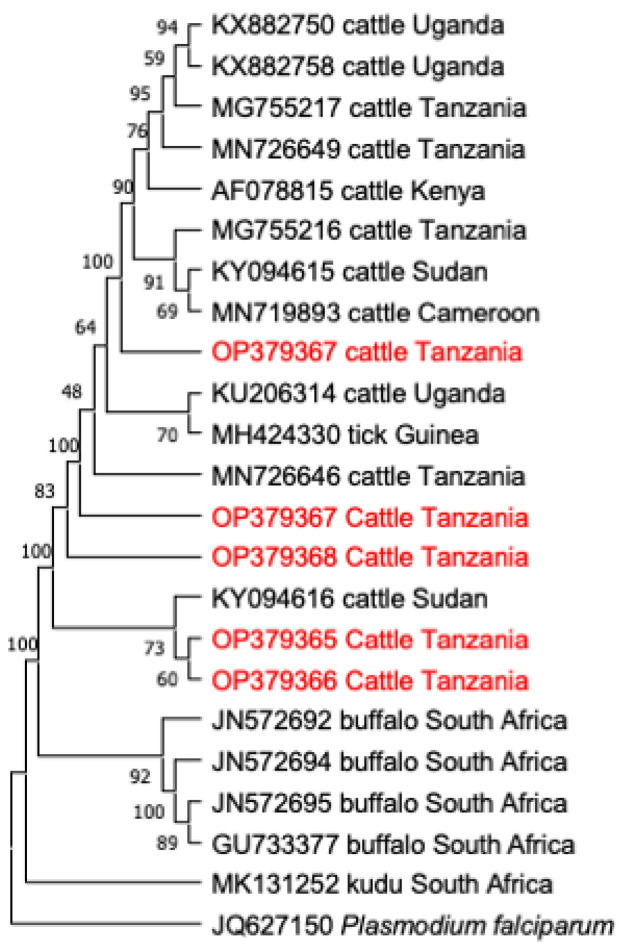
The phylogenetic analysis of *Theileria mutans* identified in this study, based on the (*18S rRNA*) gene. The tree was constructed by MEGA version 11 using the maximum likelihood method, the confidence of occurrence of the nodes was assessed by bootstrap in 1000 replications. The sequences of this study are shown in red. *Plasmodium falciparum* (JQ627150) was used as an outgroup.

**Table 1 animals-12-03171-t001:** List of primers used in the assays.

Target Gene	Assays	Primer Sequences		Annealing	
		Forward 5′ 🠖 3′ Reverse		temp. (°C)	References
*Theileria* spp. (*18S rRNA*)	PCR	GAAACGGCTACCACATCT	AGTTTCCCCGTGTTGAGT	55	[31]
	nPCR	TTAAACCTCTTCCAGAGT	TCAGCCTTGCGACCATAC	55	
*B. bigemina* (*BbigRAP-1a*)	PCR	GAGTCTGCCAAATCCTTAC	TCCTCTACAGCTGCTTCG	55	[32]
	nPCR	AGCTTGCTTTCACAACTCGCC	TTGGTGCTTTGACCGACGACAT	55	
*B. bovis* (*BboSBP-2*)	PCR	CTGGAAGTGGATCTCATGCAACC	TCACGAGCACTCTACGGCTTTGCAG	64	[33]
	nPCR	GAATCTAGGCATATAAGGCAT	ATCCCCTCCTAAGGTTGGCTAC	58	
*T. parva* (*p104*)	PCR	ATTTAAGGAACCTGACGTGACTGC	TAAGATGCCGACTATTAATGACACC	65	[34]
	nPCR	GGCCAAGGTCTCCTTCAGATTACG	TGGGTGTGTTTCCTCGTCATCTGC	60	
*A. marginale* (*groEL*)	PCR	GACTACCACATGCTCCATACTGACTG	GACGTCCACAACTACTGCATTCAAG	74–65	[35]
	nPCR	GTCTGAAGATGAGATTGCACAGGTTG	CCTTTGATGCCGTCCAGAGATGCA	74–68	
*E. ruminantium* (*pcs20*)	PCR	ACTAGTAGAAATTGCACAATCYAT	RCTDGCWGCTTTYTGTTCAGCTAK	61	[36]
		ACTAGTAGAAATTGCACAATCYAT	AACTTGGWGCRRGDARTCCTT	61	
*T. mutans* (*18S rRNA*)	PCR	GACACAGGGAGGTAGTGACAAG	CTAAGAATTTCACCTCTGACAGT	60	[37]
	nPCR	GACACAGGGAGGTAGTGACAAG	AACATTCGGAGACGCAAGCGAG	68	
*T. taurotragi* (*18S rRNA*)	PCR	GACACAGGGAGGTAGTGACAAG	CTAAGAATTTCACCTCTGACAGT	60	[37]
	nPCR	GACACAGGGAGGTAGTGACAAG	GAACCGTCCGAAAAAAGCCACG	68	

**Table 2 animals-12-03171-t002:** Tick-borne pathogens detected from Handeni, Tanga.

	Kwamsisi	Kabuku	Kwamatuku	Kwachaga	Komkonga	Total No.
**Single infection**	(*n* = 51)	(*n* = 50)	(*n* = 50)	(*n* = 50)	(*n* = 49)	positive (%)
*T. parva*	2 (4%)	3 (6%)	2 (4%)	3 (6%)	5 (10%)	15 (6)
*T. mutans*	14 (28%)	11 (22%)	7 (14%)	5 (10%)	9 (18%)	46 (18)
*T. taurotragi*	1 (2%)	2 (4%)	4 (8%)	0 (0%)	2 (4%)	9 (4)
*B. bigemina*	1 (2%)	3 (6%)	0 (0%)	3 (6%)	6 (12%)	13 (5)
*A. marginale*	3 (6%)	1 (2%)	9 (18%)	9 (18%)	1 (2%)	23 (9)
SUB TOTAL	21 (41%)	20 (40%)	22 (44%)	20 (40%)	23 (47%)	106 (42)
**Double infections**						
*T. parva* + *T. mutans*	3 (6%)	5 (10%)	2 (4%)	4 (8%)	3 (6%)	17 (7)
*T. parva* + *T. taurotragi*	0 (0%)	1 (2%)	0 (0%)	1 (2%)	2 (4%)	4 (2)
*T. parva* + *B. bigemina*	0 (0%)	0 (0%)	0 (0%)	1 (2%)	0 (0%)	1 (0)
*T. parva* + *A. marginale*	4 (8%)	2 (4%)	0 (0%)	2 (4%)	4 (8%)	12 (5)
*T. mutans* + *T. taurotragi*	5 (10%)	1 (2%)	5 (10%)	1 (2%)	2 (4%)	14 (6)
*T mutans* + *B. bigemina*	2 (4%)	2 (4%)	0 (0%)	2 (4%)	2 (4%)	8 (3)
*T. mutans* + *A. marginale*	2 (4%)	4 (8%)	5 (10%)	4 (8%)	3 (6%)	18 (7)
*T. taurotragi* + *B. bigemina*	2 (4%)	0 (0%)	0 (0%)	1 (2%)	0 (0%)	3 (1)
*T. taurotragi* + *A. marginale*	0 (0%)	2 (4%)	4 (8%)	5 (10%)	1 (2%)	12 (5)
*B. bigemina* + *A. marginale*	0 (0%)	0 (0%)	0 (0%)	0 (0%)	0 (0%)	0 (0)
SUB TOTAL	18 (35%)	17 (34%)	16 (32%)	21 (42%)	17 (35%)	89 (36)
**Triple infections**						
*T. parva* + *T. mutans* + *T. taurotragi*	0 (0%)	0 (0%)	1 (1%)	2 (4%)	1 (2%)	4 (2)
*T. parva* + *B. bigemina* + *A. marginale*	1 (2%)	0 (0%)	0 (0%)	0 (0%)	0 (0%)	1 (0)
*T. parva* + *T. mutans* + *B. bigemina*	1 (2%)	1 (2%)	0 (0%)	0 (0%)	1 (2%)	3 (1)
*T. parva* + *T. taurotragi* + *A. marginale*	1 (2%)	0 (0%)	1 (2%)	1 (2%)	1 (1%)	4 (2)
*T. mutans* + *B. bigemina* + *A. marginale*	1 (2%)	0 (0%)	1 (2%)	1 (2%)	0 (0%)	3 (1)
*T. taurotragi* + *B. bigemina* + *A. marginale*	1 (2%)	0 (0%)	0 (0%)	1 (2%)	0 (0%)	2 (1)
*T. mutans* + *T. taurotragi* + *B. bigemina*	1 (2%)	0 (0%)	0 (0%)	0 (0%)	0 (0%)	1 (0)
*T. mutans* + *T. taurotragi* + *A. marginale*	1 (2%)	0 (0%)	0 (0%)	1 (2%)	1 (2%)	3 (1)
SUB TOTAL	7 (14%)	1 (2%)	3 (6%)	6 (12%)	4 (8%)	21 (8)
**Quadruple infections**						
*T. parva* + *T. mutans* + *T. taurotragi* + *A. marginale*	1 (2%)	1 (2%)	0 (0%)	0 (0%)	0 (0%)	2 (1)
**Total positive samples**	47 (92%)	39 (78%)	41 (82%)	47 (94%)	44 (90%)	218 (87)

**Table 3 animals-12-03171-t003:** Overall prevalence of the detected pathogens.

Pathogen	Kwamsisi (*n* = 51)	Kabuku (*n* = 50)	Kwamatuku (*n* = 50)	Kwachaga (*n* = 50)	Komkonga (*n* = 49)	Overall (*n* = 250)
*Theileria parva*	14 (27.5%)	16 (32%)	5 (10%)	18 (36%)	11 (22.5%)	64 (25.6%)
*Theileria mutans*	30 (58.8%)	28 (56%)	20 (40%)	22 (44%)	20 (40.8%)	120 (48%)
*Theileria Taurotragi*	11 (21.6%)	8 (16%)	14 (28%)	13 (26%)	6 (12.3%)	52 (20.8%)
*Babesia bigemina*	8 (15.7%)	7 (14%)	1 (2%)	7 (14%)	10 (20.4%)	33 (13.2%)
*Anaplasma marginale*	15 (29.4%)	13 (26%)	20 (40%)	26 (52%)	7 (14.3%)	81 (32.4%)

**Table 4 animals-12-03171-t004:** The prevalence of tick-borne pathogens detected, based on the animal’s sex.

Sex	*T. parva*	*T. mutans*	*T. taurotragi*	*B. bigemina*	*A. marginale*
Male (*n* = 66)	16 (24.2%)	37 (56.1%)	9 (13.6%)	11 (16.7%)	20 (30.3%)
Female (*n* = 184)	48 (26.1%)	83 (45.1%)	43 (23.4%)	22 (11.9%)	61 (33.2%)

**Table 5 animals-12-03171-t005:** Observed co-infection, frequencies, number of species combination and pathogens involved.

Co-Infections	Frequencies	%	Species Combination	Pathogens Involved
Double	89	79.4	10	20
Triple	21	18.8	8	24
Quadruple	2	1.8	1	4
Overall	112	100	19	48

## Data Availability

The data presented in this study are available upon request from the corresponding author.

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
