# Peer review of "Molecular Investigation of Tick-Borne Haemoparasites Isolated from Indigenous Zebu Cattle in the Tanga Region, Tanzania"

_animals, 2022, doi:10.3390/ani12223171_

Round 1
Reviewer 2 Report
1.Discussion: The above are all analyzed factors affecting infection. Phylogenetic tree analysis showed that T. perva p104 gene sequences in this study clustered in a clade, indicating that p104 gene was conserved in the sample cattle.
Such a discussion is not sufficient.
2. Data in Table 2 is incorrect.
3. The data is about prevalence rate and mixed infection rate obtained by epidemiological analysis. However this manuscript is mainly to compare the results of this test with the gene sequences of pathogens previously detected in local or neighboring countries. The analysis of genetic variation, homology and species relationship should be emphasized in the discussion.
4. Figure 4 and 5 phylogenetic trees did not have outgroups.
5. The conclusion lacks the description of population identification and population genetic structure based on phylogenetic tree.
Round 2
Reviewer 2 Report
The research of this paper is very meaningful, and the content is also substantial. The article needs to be slightly revised to make it more suitable for publication. In addition, it is recommended that the presentation of data and other results be aesthetically pleasing for easy reading and understanding, so as to attract as many readers as possible to understand the dangers of Tick-transmitted diseases.